# Cotton-Based Rotation, Intercropping, and Alternate Intercropping Increase Yields by Improving Root–Shoot Relations

**Qingqing Lv** [1,2], **Baojie Chi** [3] , **Ning He** [1,2], **Dongmei Zhang** [2], **Jianlong Dai** [2,*], **Yongjiang Zhang** [1,*]
**and Hezhong Dong** [1,2,*]

1   State Key Laboratory of North China Crop Improvement and Regulation/Key Laboratory of Crop Growth
    Regulation of Hebei Province, College of Agronomy, Hebei Agricultural University, Baoding 071000, China
2   Institute of Industrial Crops, Shandong Academy of Agricultural Sciences, Jinan 250100, China
3   Shandong College of Agriculture and Engineering, Jinan 250100, China
*   Correspondence: daijianlong0805@126.com (J.D.); yongjiangzh@sina.com (Y.Z.); donghezhong@163.com (H.D.);
    Tel.: +86-139-6917-3767 (J.D.); +86-312-752-8108 (Y.Z.); +86-531-6665-9255 (H.D.)

**Abstract:** Crop rotation and intercropping are important ways to increase agricultural resource utilization efficiency and crop productivity. Alternate intercropping, or transposition intercropping, is a new intercropping pattern in which two crops are intercropped in a wide strip with planting positions switched annually on the same land. Transposition intercropping combines intercropping and rotation and thus performs better than either practice alone. Compared with traditional intercropping or rotation, it can increase yield and net return by 17–21% and 10–23%, respectively, and the land equivalent ratio (LER) by 20% to 30%. In crop growth and development, a balanced root–shoot relation is essential to obtain satisfactory yields and yield quality. Intercropping, rotation, or the combination can alter the original root–shoot relation by changing the ecology and physiology of both root and shoot to achieve a rebalancing of the relation. The crop yield and yield quality are thus regulated by the root–shoot interactions and the resulting rebalancing. The review examines the effects of above- and belowground interactions and rebalancing of root–shoot relations on crop yields under cotton-based intercropping, rotation, and particularly alternate intercropping with the practices combined. The importance of signaling in regulating the rebalancing of root–shoot relations under intercropping, rotation, and the combination was also explored as a possible focus of future research on intercropping and rotation.

**Keywords:** alternate intercropping; ecophysiology; root–shoot signaling; yield formation





## 1. Introduction

In China, the per capita cultivated land area is small, and the conflict between different crops has always been prominent [1]. In recent years, unreasonable farming management has led to an imbalance in the structure of agriculture [2,3]. Long-term single cultivation has been found to cause serious deterioration of land quality [4], inhibited crop growth and development, decreased accumulation of organic matter in the soil [5], decreased photosynthetic production [5,6], increased incidence of pests and diseases [7,8], and thus reduced the crop productivity [4].

Crop rotation and intercropping are used worldwide to improve crop productivity in sustainable agriculture [9]. Rotation has been demonstrated to effectively reduce constraints and increase crop yield [10]. However, the traditional rotation is not attractive because most smallholder farmers in the Yellow River Valley of China prefer to harvest the two cash crops in the same year as a result of economic considerations. Although traditional intercropping can meet farmers' requirement of harvesting two crops in one year, there still exist continuous cropping constraints [11]. Moreover, traditional intercropping is not amenable to mechanization, which is also an important reason for its low adoption in recent years [12].

To address this concern, alternate intercropping, combining rotation and intercropping as a new cropping pattern, has recently attracted wide interest [1]. It is characterized by wide-strip intercropping and interannual crop transposition (Figure 1), which can satisfy farmers' need to harvest two crops a year, reduce continuous cropping constraints, and produce better yields than traditional rotation and intercropping alone [13,14].

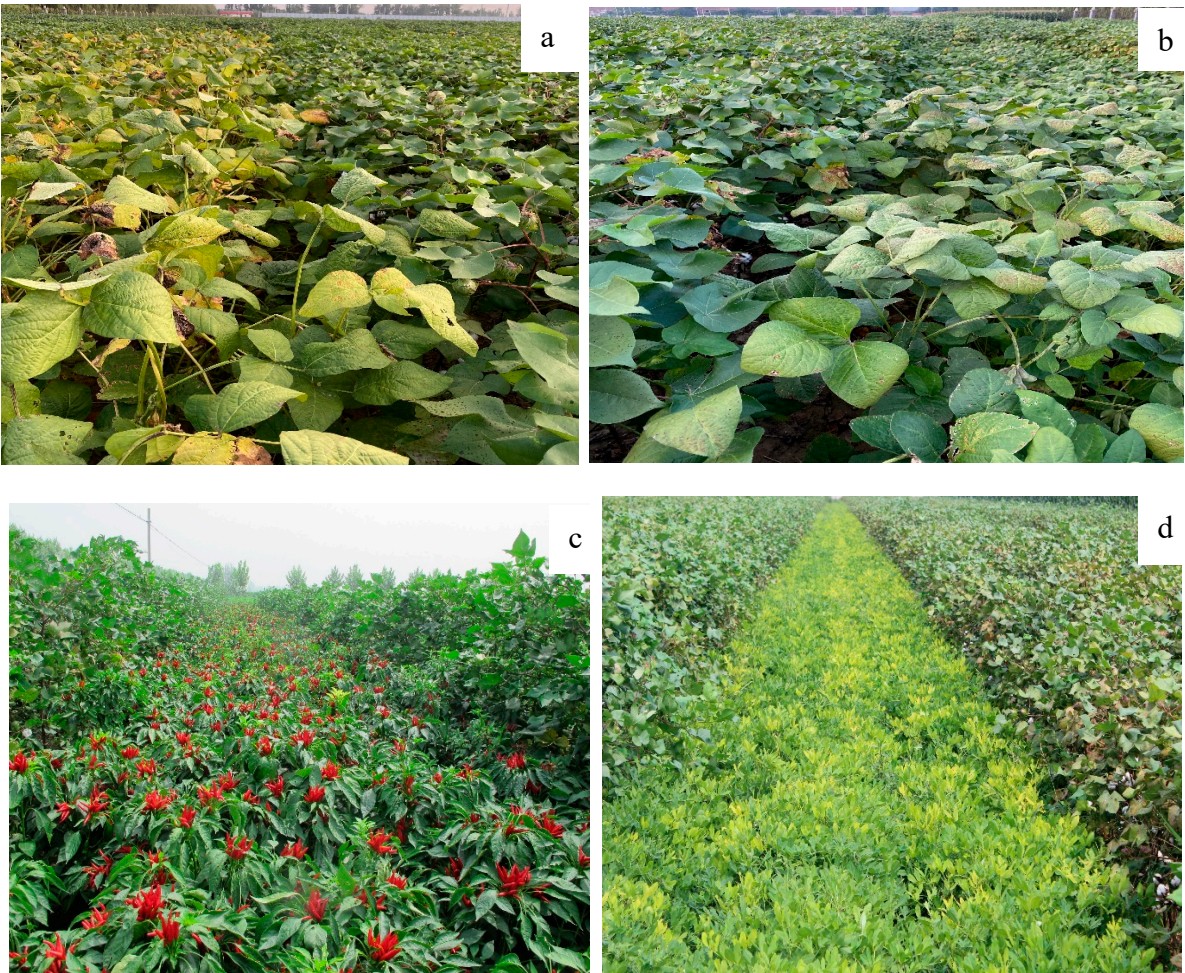

**Figure 1.** Traditional (**a**) and alternate (**b**) intercropping of cotton and soybean; intercropping of cotton and chili (**c**) and cotton and peanut (**d**).

Previous studies mainly focus on intercropping or rotation, especially on aboveground growth and development, crop yields, and benefits [15,16]. However, recently, attention has increasingly focused not only on traditional intercropping or rotation, but also on alternate intercropping, with a particular focus on the belowground, and its interactions with the aboveground [14,17]. In this review, on the basis of the literature and relevant research of the authors, we discuss how alternate intercropping affects root–shoot interactions and the rebalancing of root–shoot relations in order to provide a reference for the sustainable development of cotton intercropping and multiple cropping.

The Web of Science and Cnki (www.cnki.net; accessed on 1 January 2023) were used in a search of the literature published since 1980, with the literature search terms "Rotation" and "Intercropping". A total of 49,598 articles related to intercropping and rotation were identified, including 14,982 related to intercropping, 34,610 related to rotation, and 12 related to alternate intercropping. Among the articles, 1979 were cotton-based, including 1054 on rotation, 920 on intercropping, and 5 on alternate intercropping. Compared with single cropping, 78.2% of articles on cotton-based intercropping showed an increase in yield; compared with continuous cropping, 86.2% of articles on rotation showed an increase

in yield; and compared with traditional intercropping and rotation, 100% of articles on alternate intercropping showed an increase in crop yield.

## 2. How Cotton-Based Rotation Improves Productivity

Cotton production occurs on five continents under diverse agroclimatic conditions with contrasting productivity levels and production constraints [18]. Developing eco-efficient cropping systems in cotton fields that are productive and sustainable is key in increasing productivity and economic benefits. Rotation is a planned sequence of multiple crop species grown in recurring succession on the same area of land [17]. Cotton-based rotation is an important way to increase productivity in agro-ecosystems.

### 2.1. Productivity and Economic Benefits

Cotton-based rotation systems can improve crop yields across a broad range of environments. For example, compared with continuous cropping of cotton, rotation cropping increased cotton yields by 2.6~4.5% [15]. Rotation also promotes cotton vegetative and reproductive growth and increases plant height; numbers of true leaves, fruit branches, and squares and bolls; and boll weight [15]. In a more recent study using satellite remote sensing, the lint yield of cotton rotated with rice was greater than that of continuous cotton [19].

Cotton-based rotations also increase profitability, a key factor influencing farmer decisions to adopt a crop rotation system [20,21]. Cotton→wheat rotation produces gross margins 25% higher than those of continuous cropping because of the lower cost of production [22]. Legumes are a viable option for small landholders using cotton-based cropping in India [23], and the gross profits of this system have always been higher than those of continuous cotton [24]. Cotton→legume rotations have been widely adopted in Australia and India because of increased profitability [23,24]. The inclusion of vetch in continuous cotton or cotton→wheat system increases gross profit margins by 23% and 12%, respectively [24,25]. Thus, productivity and economic benefits are greater with cotton-based rotations than with continuous cotton, especially when rotated with legumes.

### 2.2. Soil Improvement and Root Growth

Soil physical properties such as aggregation and stability are key factors in regulating soil functions [17,26]. The benefits of crop rotation on soil physical properties are widely reported [27–29]. For example, when legumes are introduced in a crop rotation, macroaggregate formation increases because of deep roots, leaf drop, increased rhizosphere activity, and root exudates [30]. Moreover, crop rotation with legumes increases the diversity of crop species, which can reduce the need for traditional cultivation measures [17].

Crop rotation results in changes in root structure and biomass [31–33] and also causes variation in root exudates and rhizosphere microorganisms [34,35]. Roots play an important role in increasing soil organic carbon (SOC) levels, but the root traits that impact SOC likely vary widely among cover crop species [33]. Amsili et al. [36] found rotation improved the quantity, quality, and spatial distribution of roots, and changed the root-to-shoot (R:S) ratio, compared to single planting. Crop rotation also increased root cumulative carbon by 37%~46% and reduced the root carbon-to-nitrogen (C:N) ratio. Incorporating bahiagrass into the traditional peanut and cotton cropping system resulted in improved cotton root development including larger total root area, length, and biomass [32]. Rotation with sod improved the taproot system of cotton, which enables the cotton to extract nutrients and soil moisture from the deeper soil profile [32].

A meta-analysis showed that crop rotations increase microbial biomass by 21% [37]. Changes in microbial community structure are also associated with differences in rotational diversity [38–40]. Rotational diversity increases microbial community diversity and the relative abundance of fungi compared with that of bacteria [41,42]. Such changes are important because microbial community diversity is linked to functional resilience and resistance to disturbance [26]. The rotation of cotton and leguminous crops results in substantial

changes in the rhizosphere [26,35]. The rotation increased rhizosphere microbial biomass by 21% compared to continuous cropping, improved soil microbial diversity and ecological functions [37], and ultimately improved soil organic matter and soil fertility [17,29].

*2.3. Resource Utilization*

Appropriate rotation increases productivity and economic benefits [21], which is largely attributed to improved resource utilization [27,43]. Therefore, it is important to understand how previous crops affect the resource utilization efficiency of subsequent crops [17]. In cotton-based rotation systems, the advantage of rotation over continuous cropping is the effective use of available resources, such as solar radiation, soil nutrients, and water [31,44,45]. For example, in a cotton→spring wheat →reseeded→feed rape rotation system, the net photosynthetic rate of cotton increases by 44.88% to 50.37% and light energy use efficiency increases by 18.82% to 59.17% compared with those of continuous cropping [46]. Canopy apparent photosynthesis also increases under rotation, and increases in leaf area index and biomass of cotton indicate increases in cotton growth and development [44].

*2.4. Root–Shoot Interaction*

It is believed that the belowground and aboveground parts interact within a cotton-based rotation system. Interannual crop rotation resulted in changes in soil organic matter content and rhizosphere microorganisms, which would affect the growth and development of the aboveground parts. The aboveground alteration would affect root growth and development, indicating an interaction between shoot and root [26,34]. In cotton-based rotation systems, cotton is a straight-root crop with a deep root system that absorbs water and nutrients from deep soil layers, whereas wheat and corn are fibrous root crops with shallow root systems that make full use of surface soil nutrients compared with cotton [47]. Thus, a rotation of cotton and wheat (maize) has complementary advantages that maximize the nutrient utilization efficiency of each soil layer [48,49]. Compared with continuous cotton cropping, cotton-based rotation increases the water use efficiency of cotton by 28.01% to 68.35% during the late flowering period [46]. A cotton→corn→soybean rotation system consumes less fertilizer, especially nitrogen fertilizer, than continuous cotton because of the nitrogen fixation of soybean [50,51]. In a cotton→peanut rotation system, root diameter, root area, root length, and root biomass of cotton increased significantly [32,33], which enhanced the uptake of nutrients and finally improved the aboveground leaf area index and plant height traits [33], realizing the regulation of root to shoot.

*2.5. Pest and Disease Control*

Continuous monoculture allows pathogens to continue life cycles without interruption, resulting in rapid multiplication of pathogens and increases in disease severity [52,53]. By contrast, when several different crop species are planted, pathogens with a relatively narrow host range or without long-term survival capacity or dispersal do not survive in the absence of a suitable host [54]. Therefore, crop rotation can also be regarded as an important strategy to control plant diseases. Diverse crop rotations increase crop health and decrease pest occurrence [17], thus reducing production risk when compared with that in monoculture [55,56]. In a cotton→grain crop rotation system, abundances of adult *Propylea japonica* lady beetles increased, which led to reductions in aphids in center cotton plots [56]. In addition, diseases (especially rust), weeds (especially wild oats), and pests (especially wheat thrips) were 2 to 3 times less frequent in grain planted after cotton, and spiders were 3 to 4 times less abundant in cotton planted after grain than in continuous cropping [57]. The studies suggest that crop diversity in rotation ecosystems is beneficial because ecological control of pests allows reductions in pesticide inputs [17,52], especially under agricultural intensification [56].

Of course, inappropriate rotation may cause problems in crop production. Generally, the rotation of similar crops is far lower than that of different categories of crops in yield

and economic benefits. For example, the yield and economic benefits of rotation between legume crops and grain crops are better than those of rotation between different legume crops or different grain crops [58].

## 3. Cotton-based Intercropping

Intercropping is a method of planting at least two crops in the same season in rows or bands [59]. Cotton-based intercropping is also an important way to increase productivity in agro-ecosystems. In a global meta-analysis, compared with monocultures under the same management, both low- and high-yield intercropping strategies saved land by 16% to 29% and reduced fertilizer use by 19% to 36% [16].

### 3.1. Productivity and Economic Benefits

Cotton-based intercropping systems are considered a promising strategy in sustainable cotton production, particularly for small landholdings [60]. Cotton is suitable for intercropping because of its wide row spacing, slow growth in the initial stage, and relatively long period of growth and development [61]. Vacant space between rows of a cotton crop allows two to three months to grow a short-duration intercropping. Legumes, including gram, bean, and cowpea, grow rapidly and complete life cycles in a short time, which is highly suitable for intercropping with cotton [61]. Although yields of intercropped cotton can be 8% to 31% lower than those of monoculture cotton, total productivity and net income of intercropping systems are much higher than those of monoculture [62]. Farm income under different cotton-based intercropping systems increases by 30% to 40% [63]. However, in the case of traditional narrow row spacing, the effects of intercropping are not good. Therefore, cotton must be planted with wide row spacing to ensure sufficient space for intercropping [64].

### 3.2. Rhizosphere Microbial Community

Exchanges and competition for mineral nutrients and water, soil microorganisms, and other resources are primarily concentrated belowground, particularly in the rhizosphere [65]. Intercropping not only affects root growth, morphology, and function, but also alters the rhizosphere microbial community, promotes or inhibits the production of plant root exudates [66], and leads to interactions among roots of intercropped crops. Plants and rhizosphere bacteria interact closely in the rhizosphere [67]. Crop plants can stimulate rhizosphere bacteria by secreting root exudates and metabolites and thus alter interactions between rhizosphere bacterial communities and plants [68].

### 3.3. Resource Utilization

Intercropping greatly increases net returns compared with those from a single crop, which is largely attributed to improved resource utilization [16]. Intercropping has wider and more rapidly developing coverage than that in monoculture, which increases the interception of solar radiation and thus increases the effective use of sunlight [69–71]. In intercropping systems, the two intercropped crops have asynchronous canopy patterns and different maturity dates [69]. Therefore, because of the expansion of leaf area throughout the growing season, light interception may increase by 30% to 40% compared with that in monoculture [72]. The microclimate within the canopy of intercrops regulates temperature extremes, and in the summer, a widespread canopy of a main crop such as cotton reduces temperature and air movement, leading to decreased evaporation loss and increased relative humidity [73]. For example, compared with single cropping, jujube–cotton intercropping decreases soil temperature under jujube and cotton by 0.04 °C to 0.87 °C and 0.63 °C to 2.92 °C, respectively, throughout the growth period, except in April and May, and decreases transpiration and soil evaporation in intercropped jujube by 12 mm and 39 mm, respectively [74].

Interspecific exchanges, competition, and other interactions occur in cotton-based intercropping systems, which can significantly improve soil microbial activity [75] and

increase the decomposition of humus and the transformation of organic matter and nutrients [76]. Long-term (10 to 16 years) experiments in soils of different fertility show that the average grain yield of intercropping systems is 22% higher than that of monocultures, and yields are more consistent year to year [77]. A possible explanation for why intercropped crops can absorb more nutrients is that the roots of intercropped crops are more developed and have longer-lasting functions than those of monocropping crops [14]. Compared with traditional cotton monocropping, cotton–halophyte intercropping increases root mass and density at the 0–20 cm soil depth [78], and cotton–mung bean intercropping increases total land output by 16.6% to 19.8%, total nitrogen uptake by 27.9% to 45.3%, water use efficiency by 17.0% to 36.3%, and economic benefits by 31.7% to 51.9% [79].

*3.4. Pest Control*

Intercropping is important in sustainable cotton production because it helps to reduce populations of insect pests by attracting natural enemies and typically produces stable yields and high profits [80]. Arthropod community structure is associated with the spatial and temporal structure of crops [53,81]. The increase in vegetation diversity in intercropped cotton fields alters the temporal and spatial pattern of crops and thus improves the stability of arthropod communities [82,83]. Cotton–cowpea intercropping reduces the number of apterous *Aphis gossypii* per cotton plant by 31% to 43% compared with that in monocropping [84]. Intercropping cotton with trap crops such as corn, alfalfa, mung bean, and cowpea can effectively trap and reduce pest abundance on cotton [85–87]. The loss rate of diseases of intercropped cotton and corn is lower than that of monocropped cotton [88]. It was found that using strip intercropping with cotton was effective in controlling early leaf spots of groundnut, which also reduced fungicide application [89].

In addition, vegetation structure, ventilation, humidity, and temperature differ between cotton intercropping ecosystems and single-plant ecosystems [73,74]. This microclimate change can increase the number of natural enemies on the one hand, reduce the number of pests on the other hand, and finally reduce pest damage [90,91]. Crops with plant height differing from that of cotton, such as corn or fruit trees, greatly affect temperature, humidity, wind speed, and light in intercropping systems. Therefore, climate factors directly or indirectly affect the development and survival of cotton pests, thus affecting population densities [92–94].

## 4. How Alternate Intercropping Improves Crop Productivity

Alternate intercropping, or transposition intercropping, is a new intercropping mode in which two crops are intercropped in a wide strip with planting positions switched annually on the same land [95]. Transposition intercropping effectively combines rotation with intercropping to realize the benefits of increases in yields from intercropping and increases in efficiency from rotation [95–97]. Currently, transposition intercropping is used in cotton, corn, legumes (peanut, soybean, and mung bean), and other crops and has achieved good results, showing broad prospects for increases in application [96,97].

*4.1. Productivity and Economic Benefits*

Intercropping can satisfy the need for farmers to harvest two crops per year [77], and rotation can reduce constraints associated with continuous cropping [17]. Therefore, when intercropping and rotation of different crops are combined, farmers can harvest two crops within one year, reduce problems associated with continuous cropping, and further improve crop productivity and land use efficiency [16,77]. The average land equivalent ratio can increase by 20% to 30% compared with that in traditional intercropping [95–97]. Traditional cotton–peanut intercropping increases seed cotton yield by 16.9% and decreases peanut yield by 5.6%, whereas alternate intercropping of cotton and peanut increases cotton yield by 21% without sacrificing peanut yield. Therefore, the crop output value under alternate intercropping was 4.5% higher than that under traditional intercropping, and the net return exceeded that under traditional intercropping by 10% [1]. In maize and

peanut alternate intercropping, annual yields of maize and peanuts increased by 19.68% and 17.29% and net revenues increased by 23.14% and 13.99%, respectively, compared with those in traditional intercropping [95]. Thus, compared with traditional intercropping, alternate intercropping has been shown to increase crop productivity and economic return without additional inputs [95,96], indicating it is a promising cropping system.

*4.2. Soil and Rhizosphere Microbial Community*

Alternate intercropping has been shown to have important effects on soil microbial community structure and function [26,34] which may be associated with increases in nutrient uptake [44,50] and water use efficiency, as well as increases in dry matter accumulation and N translocation [31,45]. Compared with continuous cropping of maize and peanut, cotton–peanut alternate intercropping increases soil carbon stocks in the 10–20-cm soil layer by 20.11% (maize) and 34.19% (peanut) [95], and as a result, soil erosion and carbon emissions decrease [98]. Thus, alternate intercropping may increase crop yields by improving soil properties [96,97], with increases in soil organic carbon levels, decreases in carbon mineralization rates, and increases in nutrient availability [95].

Both rotation and intercropping directly affect soil and rhizosphere microbial communities [26,68]. Compared with monoculture, diversified planting can significantly increase the diversity and composition of bacterial communities, with consequent effects on ecosystem functioning [99,100]. Related analysis shows that alternate intercropping may increase microbial richness [95,101] and dramatically alters the abundance and composition of soil bacteria in the topsoil of both crops of a strip [102]. Therefore, increases in microbial richness and diversity and improvements in microbial community structure may be other reasons for the high productivity in alternate intercropping.

*4.3. Resource Utilization*

Alternate intercropping may provide distinct advantages above ground [95]. Well-designed alternate intercropping can optimize the canopy microclimate and thus increase aboveground photosynthetic production and assimilate partitioning [69,71,73,103]. Alternate intercropping of cotton and peanut increases the uptake of nitrogen, phosphorus, and potassium by 6.3, 11.5, and 7.3%, respectively. Net photosynthetic rate, chlorophyll content, and maximum leaf area index of peanut increase by 7.2, 8.9, and 4.4%, respectively [1]. Strip width can also significantly affect light capture and light use efficiency of intercropped crops [69]. Therefore, appropriately increasing the width of strips to alleviate negative effects on shorter crops in an intercropping system can effectively increase total productivity and economic benefits [95].

Wang et al. [104] found that variation in the proportion of border rows is due to changes in strip width between 1 and 4 m. Therefore, when the strip width is increased to 5 m, the adverse effects of shading on the yields of shorter crops are eliminated. For example, under maize and peanut alternative intercropping, compared with continuous cropping, intercropping increases the value of soil plant analysis development (SPAD), net photosynthetic rate, and the dry weight accumulation of maize, and the reduction in effects of shading by wider strips and the alleviation of continuous cropping obstacles by rotation increase the SPAD value and dry weight accumulation of peanut, which may explain the high productivity in alternate intercropping [95].

*4.4. Interactions under Alternate Intercropping*

4.4.1. Three Types of Interactions

Plant root and shoot form a unit in which each part depends on and promotes and restricts the other [105]. Under intercropping or rotation, microecological conditions change above- and belowground parts [106,107], and the original root–shoot relation is altered. Therefore, an inevitable series of changes occur to "rebalance" the root–shoot relation. In the process of rebalancing, yield generally decreases when roots and stems are antagonistic to one another, whereas yield increases when roots and stems are synergetic [95,101].

Root–shoot interactions and the rebalance within a crop plant are also affected by the intercropped crop. Specifically, root–root and shoot–shoot interactions occur between different crop species under intercropping [108]. Therefore, different types of interactions occur under alternate intercropping of cotton and leguminous crops [101,102]. Three types of interactions under alternative intercropping of cotton and legumes can be identified, namely "aboveground (shoot–shoot) interactions", "belowground (root–root) interactions", and "root–shoot interactions". Belowground interactions regulate nitrogen, phosphorus, and potassium uptake by crops [59,103,109]. Aboveground interactions regulate the canopy structure of both crops [110,111], and interactions between roots and shoots modulate the rebalancing of root–shoot relations. The overall effects of the different interactions in alternate intercropping of cotton and soybean are shown in Figure 2.

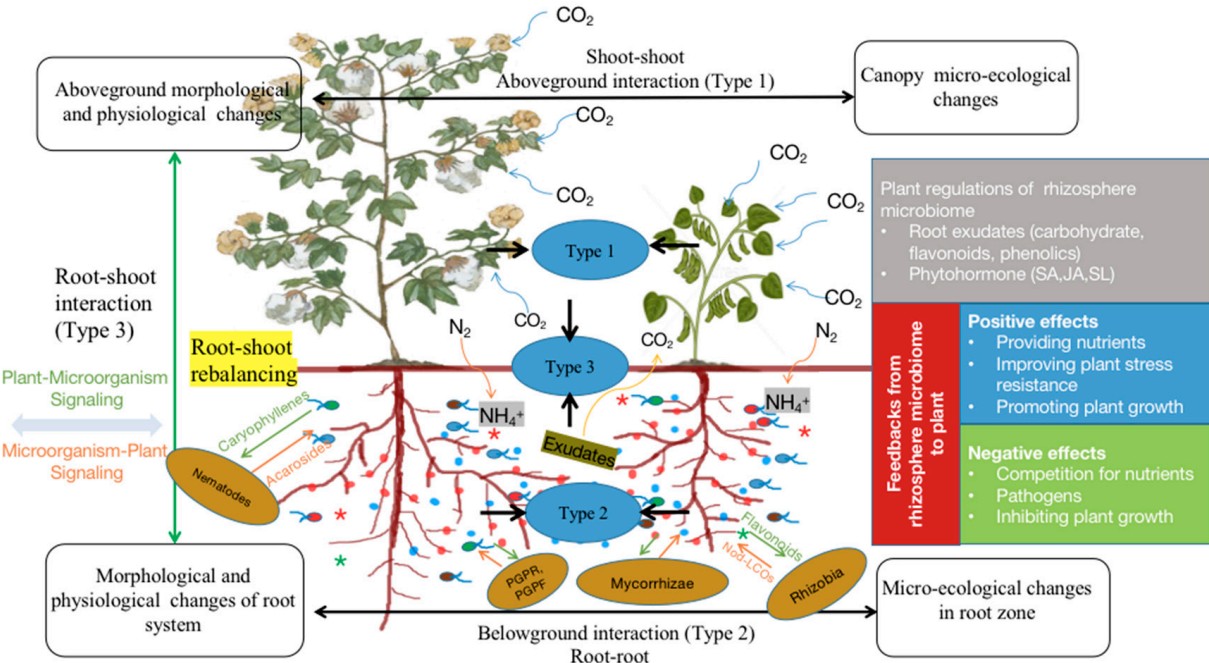

**Figure 2.** Schematic diagram of rebalancing root–shoot relations in alternate intercropping with cotton and soybean (Refer to Liu et al., 2021 [112] for the text in the box in the figure).

Under alternate intercropping, interspecific aboveground interactions (Type 1) change canopy micro-ecology (e.g., temperature, humidity, and $CO_2$ concentration), resulting in morphological and physiological alterations of aboveground parts. Interspecific belowground interactions (Type 2) change the micro-ecology of root zones, especially by affecting rhizosphere microorganisms, which results in changes in root morphology and physiology. Intraspecific root–shoot interactions (Type 3) are also mediated by interspecific interactions (Types 1 and 2) under alternate intercropping, which leads to "rebalancing" of the root–shoot relation and ultimately regulates the formation of crop yield and quality. Signal transmission is involved in the three types of interactions.

### 4.4.2. Root–Shoot Interaction

In an alternate intercropping system, the soil–air interface creates a spatial division between above- and belowground interspecific interactions [103]. Root barriers can be used to separate belowground interactions from those above ground [102,109,113]. Some studies indicate that compared with single-crop plantings, root interactions are more important than shoot interactions in determining the productivity of an intercropped system [59,105], while other studies indicate that aboveground interactions have greater effects than those below ground in intercropping [106,111]. In addition, effects on the productivity of both root and shoot interspecific interactions vary according to crop species combinations [103,114] and are further modified by the availability of environmental

resources [109,111]. Therefore, it is important to achieve a thorough understanding of the effects of above- and belowground interactions on plant growth in order to increase the advantages of alternate intercropping.

Alternate intercropping optimizes root–shoot relations by combining rotation with intercropping [95]. Root absorption and photosynthetic production capacity improve synergistically [111], indicating a rebalance in the root–shoot relation, which leads to further increases in yields and benefits of intercropping systems [103,105]. With alternate intercropping of cotton and peanut, both leaf photosynthesis and root absorption of nitrogen, phosphorus, and potassium increase greatly [102]. The rhizosphere bacterial community is also significantly enriched under alternate intercropping [59,115], and there is feedback from changes in rhizosphere bacterial communities on aboveground processes [106]. As a result, alternate intercropping rebalances root–shoot relations to ultimately improve system productivity, compared with traditional rotation or intercropping [103,109]. Under alternative intercropping, soil microorganisms and canopy microclimate are changed to a greater extent than those in the traditional cropping mode [95,101]. Improvements in rhizosphere microbial community structure and function and changes in canopy microclimate affect plant nutrition [106,109], growth and development [103], and canopy photosynthesis and assimilate distribution [111]. As a result, root–shoot relations are rebalanced under alternate intercropping, leading to improvements in crop yield and quality [14,16]. Thus, rhizosphere microorganisms affect root–shoot interactions and have important roles in achieving the high yields obtained with alternate intercropping. The importance of rhizosphere microbial communities should be considered when designing sustainable cropping systems [95]. Therefore, for a monocultured or traditionally intercropped crop, there exists a balancing of root–shoot relations as a result of root–shoot interactions. However, under alternate intercropping, the original root–shoot coordination is broken by interspecific aboveground interactions and interspecific belowground interactions, resulting in a new balancing or rebalancing of the root–shoot relationship.

### 4.4.3. Signaling in Interactions

Transmission of signal molecules is important in the three types of interactions under alternate intercropping [116,117]. Signal molecules include microbial antibiotics, volatile organic compound (VOC) and quorum-sensing (QS) polypeptides [118], and small and medium organic molecules and nucleic acids that transmit information in the xylem and phloem [119]. The compounds can act as both intraspecific and interspecific signal molecules. When the aboveground part of a crop plant is subject to stress, signal substances are generated and transmitted to roots to promote nutrient absorption [120–122]. Roots then transmit absorbed nutrients to the aboveground to alleviate the stress, improve the crop growth environment, and form a balanced system of nutrient circulation [117].

In addition, signal regulation of root–shoot interactions is also reflected in the senescence of crop leaves [123]. In the premature senescence of cotton leaves caused by potassium deficiency, regulation is primarily accomplished by a root–shoot feedback signal that changes potassium absorption by roots followed by feedback to leaves [124]. Grafting untransformed (wild-type) tobacco scions onto the roots of transgenic plants that can produce many cytokinins does not prevent normal leaf senescence of the wild-type scions [125]. This result indicates that the shoot regulates the activity of roots, followed by root feedback to the shoot to rebalance the root and shoot relation [117]. According to girdling and grafting experiments, the regulation of senescence of canopy leaves by roots primarily occurs through the transport of cytokinins and other hormones as signal molecules from roots to the canopy [120,123]. Osterlund et al. [126] also found that *Arabidopsis* ELONGATED HYPOCOTYL5 (HY5), a bZIP transcription factor that regulates growth in response to light, is a shoot-to-root mobile signal that mediates light promotion of root growth and nitrate uptake. Mobile *HY5* coordinates light-responsive carbon and nitrogen metabolism and hence shoot and root growth. Therefore, signal molecules have important roles in root–shoot interactions by regulating shoot–root rebalancing. Thus, under intercropping or rotation,

but especially under alternate intercropping, signal molecules may have important roles in intraspecific root–shoot interactions and in interspecific root–root or canopy-canopy interactions. The participation and regulation of signal molecules promote nutrient rebalancing between shoots and roots.

## 5. Discussion and Conclusions

Intercropping and rotation are both effective ways to improve crop productivity and ecological benefits. Rotation can increase soil nutrient and organic matter content, change soil microbial diversity and its ecological function, affect root morphology and function, and then regulate the growth and development of aboveground canopy and the formation of yield and quality [1,17,26]. Intercropping, on the one hand, changes the root zone microecology, and affects root morphology and function through root interspecific interactions of intercropping crops; on the other hand, it changes the canopy microclimate, affects the plant type and canopy structure of crops, and regulates the aboveground photosynthetic production and assimilate partitioning and crop yield formation [14,16,59,77]. Intercropping and rotation break the original root–shoot relationship by changing the root–shoot physiological ecology and achieving a new coordination of root and shoot, that is, "rebalancing", through root–shoot interaction, and thus the crop yield and quality formation are regulated by the rebalancing [101]. The new alternate intercropping mode characterized by wide strip intercropping of two crops and rotation of annual planting positions has the functions of intercropping and rotation, which can more effectively coordinate the root–shoot relationship and promote the formation of crop yield and quality through root–shoot rebalancing (Figure 2).

Rotation and intercropping, and especially their combination in alternate intercropping, can fully utilize natural resources, improve productivity, and will play increasingly important roles in future agricultural production and ecosystems [96,97]. The research on them will also shift from focusing on the aboveground to both the above- and belowground parts, and from focusing on the interaction between the aboveground canopy to the interaction between the root zone and the root canopy. Advanced technologies and means such as molecular biology and smart agriculture are also bound to be applied to crop rotation and intercropping.

Of course, limitations and corresponding solutions regarding alternate intercropping need investigation. Under a climate change scenario, crops will suffer different abiotic stress conditions such as salt, heat, and drought. We can cultivate varieties with strong stress resistance for intercropping and rotation [127–129]. Selecting crops with strong abiotic resistance such as cotton and sorghum (*Suaeda salsa* or *Medicago sativa*) for rotation or intercropping can alleviate salinity damage [78]. Developing shade-tolerant cotton varieties and adopting cotton–fruit intercropping can alleviate the stress of high temperature on cotton to a certain extent. Under limited water or nitrogen fertilizer input, the temperature and precipitation in a year vary greatly. In this case, it is necessary to properly adjust the intercropping time, strip width, and crop variety. A crop model can be adopted to manage and mitigate the climate risk under intercropping [130]. Based on practical needs and development trends, the following aspects should be paid particular attention to in the future.

### 5.1. Modeling of Intercropping and Rotation

Crop modeling is an effective tool to manage and reduce the climate risk of planting systems [130–132]. On farmland and a certain regional scale, it can quantify the aboveground yield advantage of intercropping (rotation) as well as the distribution and utilization of belowground resources so as to achieve a wide range of application assessments and reduce the risk of climate change [110]. At present, the internationally recognized models related to intercropping mainly include the INTERCOM model, plant structure and function model (FSPM), and wheat/cotton model [131,132]. Based on the crop growth and development process, these models can be combined with intelligent agricultural

technology to achieve accurate simulation of interspecific and intraspecific interactions, crop growth and development, and yield formation in intercropping and rotation systems. Therefore, modeling of intercropping or rotation should be studied and developed according to the needs.

*5.2. Root–Shoot Signal Transmission*

It is important to deepen the study on the release and action mechanism of root exudates in the intercropping system, the dynamic process of the rhizosphere for the efficient utilization of nutrients [133], and especially the study on the microbial diversity and function of the rhizosphere [118]. It is necessary to study the signal molecules involved in the regulation of root–shoot interaction under intercropping or rotation and their mechanisms, so as to provide a theoretical basis for coordinating and balancing root–shoot relationships [117]. It is also necessary to study the role and mechanism of signal molecules in the intercropping and rotation system in the "aboveground interaction system", "belowground interaction system", and "root–shoot interaction system" under the background of global climate change so as to provide basis and support for the design of agricultural ecosystems that adapt to climate change.

*5.3. Integration of Cultivar, Agronomy, and Machinery*

Selection and utilization of appropriate crop types or combinations of cultivars can strengthen exchanges and cooperation among crops and reduce interspecific competition [114]. The operation of intercropping and rotation is much more complicated than monocropping, and sustainable development can be achieved only by mechanization. At present, a strip intercropping of corn and soybean with a land equivalent ratio of more than 1.5 has been widely demonstrated in China [16,77]. The unit yield of corn is equivalent to that of single cropping with an extra harvest of soybean, and it is thus being considered of great significance for easing the conflict between grain and oil crops for land [16]. The unit yield of corn is equivalent to the unit yield of a single crop, and there is additional soybean harvest in the system. Therefore, it is considered to be of great significance for easing the land conflict between food and oil crops. However, there are disadvantages related to mechanization and weed control. In order to give full play to the potential of increasing production and efficiency of the technology, measures should be taken to make the system realize the integration of machinery and agronomy. Intelligent agricultural technologies such as intelligence, digitization, and geospatial information technology should be introduced into the research and practice of intercropping and rotation so as to promote the sustainable development of intercropping and rotation [128].

In conclusion, intercropping and rotation affect intraspecific root–shoot interactions as well as interspecific root–root and shoot–shoot interactions. As a result, root–shoot relations are rebalanced, and crop yields and quality are regulated by the rebalancing. Alternate intercropping is a typical practice that regulates and improves yield and efficiency through three types of interactions to rebalance root–shoot relations. Future research on alternate intercropping should further reveal the interaction mechanisms of the competition and the exchange of aboveground and belowground resources. It is also necessary to further study how to match cotton with intercropping crop varieties and optimize agronomic measures such as sowing date, plant density, fertilization, and chemical regulation. With this information, the optimal composite planting scheme can be designed to effectively adjust the composite population structure, make better use of the spatial–temporal niche differences between intercropped crops, meet the special needs for light and heat resources, improve resource utilization efficiency and crop productivity, and thus provide new theoretical and technical support for the sustainable development of crop production.

**Author Contributions:** Conceptualization, H.D., J.D. and Y.Z.; writing—original draft, Q.L.; writing—review and editing, H.D., J.D. and Y.Z.; visualization, B.C., N.H. and D.Z.; funding acquisition, J.D. and Y.Z. All authors have read and agreed to the published version of the manuscript.

**Funding:** This work was financially supported by the National Natural Science Foundation of China (32101844), the National Key Research and Development Program of Hebei Province (22326403D), and the Natural Science Foundation of Shandong Province (ZR2022MC103).

**Data Availability Statement:** All data obtained through the Web of Science and Cnki (www.cnki.net).

**Acknowledgments:** We are grateful to Cundong Li and Guiyan Wang for their expert opinion during writing the manuscript.

**Conflicts of Interest:** The authors declare no conflict of interest.

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
