# Peer review of "Cotton-Based Rotation, Intercropping, and Alternate Intercropping Increase Yields by Improving Root–Shoot Relations"

_agronomy, doi:10.3390/agronomy13020413_

Round 1

Reviewer 1 Report

Specific comments

As a researcher in the field of farming, I am very interested in your work. I have looked thoroughly at your article and I see that you did a lot of work on it.

However, There are some problems in the article that need to be solved, if I understand your description correctly. Because there is no line number in the full text, the workload of review is increased. As far as I see, the paper can not be accepted if the points below are dealt with appropriately.

Abstract

1. Page 1, line 3, "mode" would be better replaced by "pattern".

2. Page 1,line 11-16.In this review, different cotton-based ......intercropping with the practices combined, these two sentences are somewhat repetitive, and suggestions can be combined to discuss.

3. Page 1, line20-21. Perhaps the keywords could be replaced to increase the chances of the article being searched.

4. Missing data in abstract.

Introduction

5. Page 2, lines 6 and 7, Suggestions to add to the deficiencies of single intercropping and crop rotation.

6. Page 2, lines 2 and 10, inconsistent hyphenation, e.g.  "wide-strip" and "rootshoot"

7. Page 2, line 10, suggests adding the effect of alternate intercropping on cotton yield, and the relationship between root system and yield.

8. Page 1-2. The introduction is a little weak, please modify.

9. Page 3, lines 11-12. It is suggested to reconsider this sentence. 

10. Page 2,lines 6-12, this paragraph can be expanded.

11. Page 2,lines 21~23 ,there is a lack of comparison of crop yields between alternating and conventional rotation.

How Cotton-based Rotation Improves Productivity

12. Page 2, line 31, in "is key", insert an a between is and key.

13. Page 2, line 31, "to increasing" is incorrect.

14. Page 2, lines 34-36, suggest putting this paragraph in introduction.

15. Page 3, reference 8 is missing from the text.

16. Page 3, lines 1-2, inconsistent grammatical tenses in the text.

17. Page 3, line 12, "cropping" should be "intercropping".

18. Page 3,line 13.“Cotton–legume rotations”The rotation symbols in this sentence should be changed and it is recommended that all the rotation in the article should be checked and modified.

19. Page 4, line 5, "change" should be "changing".

20. Page 4, lines 5-6, sentence error.

21. Page 4,lines 5-16,Crop rotation results in changes ......improved soil organic matter and soil fertility, there is little content about root system structure in this paragraph, and it is recommended to supplement

22. Page 4,lines 31-32.“because nitrogen fixation by soybean promotes  absorption of nutrients and increases yields of the three crops.”This sentence is suggested to be changed to “The nitrogen fixation of soybean can consumption so as to increase yield.”

23. Pages 3-4, sections 2.1, 2.2 and 2.3, discuss only the beneficial aspects of cotton rotations with other crops and lack some of the shortcomings of rotations.

24. Page 5,lines 5-6“Therefore,crop rotation is a fundamental strategy to control residue-borne plant diseases.”should be changed to “Crop rotation can also be regarded as a strategy to control residual plant diseases.”

25. Page 2-5. This section describes the advantages of crop rotation productivity, economic benefits, resource use, soil properties, root growth and biodiversity, but does not seem to be linked to root-shoot relationships.

26. The title is mainly related to the root cap. It is suggested to adjust the order of the subtitles and contents (The same is true of the following two parts).

Cotton-based Intercropping

27. Page 6, section 3.2, description of competitive relationships between cotton and other intercrops is lacking.

28.  Page 6, section 3.2, Intercropping improves the physicochemical properties of the soil, which is mentioned in the section on crop rotation and is not mentioned here.

29. Page 6,lines 30-32. In this sentence “Compared with traditional cotton monocropping, cotton–halophytes intercropping increases root mass and density at the 020-cm soil depth”, “020-cm” Check whether it is correct.

30. Page 7, line 1, intercropping can alter not only the root secretions of the crop but also the root structure and growth of the crop, which is not mentioned here.

31. Page 7, line 24, intercropping can change the microclimate environment and will increase the number of beneficial insects and thus reduce the number of pests, not mentioned here

32. Page 7, line 24. Note the relationship between punctuation and spaces.

33. Page 6-7, 3.2 has a lot of content, but 3.3 has little content. It is not written around the main content of the article.

How Alternate Intercropping Improves Crop Productivity

34. On page 7, "How Alternate Intercropping Improves Crop Productivity" talks about the role of "Alternate Intercropping", which is inconsistent with the text, which mostly talks about the results associated with intercropping cotton with other crops and does not cover the "Alternate Intercropping" aspect.

35. Page 7,lines 29-31.Alternate intercropping, or transposition intercropping, is a new intercropping mode in which two crops are intercropped in a wide strip with planting positions switched annually on the same land [2,84].” Check whether the references are correct.

36.  Page 8, line 12, the text mentions cotton-peanut intercropping compared to traditional intercropping and does not raise which crop the traditional intercropping is.

37. Page 8,lines 17-18.“In maize–peanut alternate intercropping. The symbol in the sentence is suggested to be deleted.

38. Page 8, section 4.1, most of the text refers to the effect of cotton intercropping with a single crop on yield and productivity, with no mention of the rotation associated with cotton intercropping systems, which is inconsistent with the title.

39. Page 9,line 6,“under maize–peanut alternative intercropping”This sentence is suggested to be changed to“under maize and peanut alternative intercropping”

40. Page 9,line 7,“SPAD”The first occurrence in the text should be the full name.

41. Page 9,lines 11-12, it is suggested to add a title between the two paragraphs

42. Page 9, line 33, format error.

43. Page 9, line 33. Please note the formatting, the space before the paragraph.

44. Page 10 line 16, it is suggested to put Figure 1 in the front to facilitate readers' understanding.

45. Page10, line 18, format error.

46. Page 10, line 23, no mention of the advantages of "alternate intercropping" in relation to "intercropping".

47. Page 10, line 18. Please note the formatting, the space before the paragraph.

48. Page 11, line 18. Please note the formatting, the space before the paragraph.

49. Page 12,line 4.Arabidopsis ELONGATED HYPOCOTYL5 (HY5)Check whether are correct.

50. Page 12 line 14, no punctuation at the end of the sentence.

Discussion and Conclusions

51. Page 12, line 20-25. Please note that the format of the references is the same as before.

52. Page 12, line 25, discussion of "alternate intercropping" is lacking.

53. Page 12, line 32, missing punctuation.

54. Page 13, line 7. Please note the correct use of punctuation at the end of sentences.

55. It is suggested to add the limitations of discovery, which can be avoided in future exploration.

56. A discussion of crop rotation patterns and years under intercropping systems in cotton is also needed and is not covered in the paper.

References

57. DOI numbers are uniformly not added, or uniformly added, and the format should be uniform.

58. Uniform abbreviations or uniform non-abbreviations for reference journals, please.

59. The format of all references cited is changed to corner mark.

60. According to the journal requirements, the journal in the reference needs to be abbreviated, so please revise the issue similar to the case of reference 2, 3, etc.

Author Response

Reply to review comments

Reviewer #1:

As a researcher in the field of farming, I am very interested in your work. I have looked thoroughly at your article and I see that you did a lot of work on it.

Reply: Thank you for your interests and comments.

However, There are some problems in the article that need to be solved, if I understand your description correctly. Because there is no line number in the full text, the workload of review is increased. As far as I see, the paper can not be accepted if the points below are dealt with appropriately.

Reply: Thank you for your comments. We are sorry for make you the inconvenience. We have taken into account of all your comments and the revised the paper accordingly.

     Abstract

  1. Page 1, line 3, "mode" would be better replaced by "pattern".

Reply: Agreed. It was replaced.

  1. Page 1,line 11-16. In this review, different cotton-based ......intercropping with the practices combined, these two sentences are somewhat repetitive, and suggestions can be combined to discuss.

Reply: Agreed. These two sentences have been modified.

  1. Page 1, line20-21. Perhaps the keywords could be replaced to increase the chances of the article being searched.

Reply: Agreed. We have replaced “rotation and intercropping” with “root-shoot signaling”.

  1. Missing data in abstract.

Reply: Important data have been added in the abstract.

 Introduction

  1. Page 2, lines 6 and 7, Suggestions to add to the deficiencies of single intercropping and crop rotation. 

Reply: Agreed. The deficiencies of single intercropping and crop rotation have been added.

  1. Page 2, lines 2 and 10, inconsistent hyphenation, e.g.  "wide-strip" and "root–shoot"

Reply: We apologize for this careless mistake. The hyphenation has been unified.

  1. Page 2, line 10, suggests adding the effect of alternate intercropping on cotton yield, and the relationship between root system and yield.

Reply: Agreed. They have been added.

  1. Page 1-2. The introduction is a little weak, please modify.

Reply: The introduction has been improved accordingly.

  1. 9. Page 2,lines 6-12, this paragraph can be expanded.

Reply: The first 3 paragraphs of the introduction has been expanded instead of expanding the suggested paragraph.

  1. Page 2,lines 21~23 ,there is a lack of comparison of crop yields between alternating and conventional rotation.

Reply: The comparison has been added.

How Cotton-based Rotation Improves Productivity

  1. Page 2, line 31, in "is key", insert an a between is and key.

Reply: “a” has been inserted.

  1. 12. Page 2, line 31, "to increasing" is incorrect.

Reply: "to increasing" has been changed to "to increase".

  1. 13. Page 3, reference 8 is missing from the text.

Reply: Reference 8 was included.

  1. Page 3, lines 1-2, inconsistent grammatical tenses in the text.

Reply: We apologize for this careless mistake. The sentence has been modified.

  1. Page 3, line 12, "cropping" should be "intercropping".

Reply: Cotton-based cropping system includs rotation and intercropping. So "cropping" seems better than "intercropping".

  1. Page 3,line 13.“Cotton–legume rotations”The rotation symbols in this sentence should be changed and it is recommended that all the rotation in the article should be checked and modified.

Reply: We apologize for this careless mistake. The rotation symbols have been modified in the text.

  1. Page 4, line 5, "change" should be "changing".

Reply: It was changed to "changes".

  1. Page 4, lines 5-6, sentence error.

Reply: The sentence has been modified.

  1. Page 4,lines 5-16,“Crop rotation results in changes ......improved soil organic matter and soil fertility”, there is little content about root system structure in this paragraph, and it is recommended to supplement.

Reply: Agreed. The discussion about root system structure has been added.

  1. Page 4,lines 31-32.“because nitrogen fixation by soybean promotes absorption of nutrients and increases yields of the three crops.”This sentence is suggested to be changed to “The nitrogen fixation of soybean can consumption so as to increase yield.”

Reply: The sentence has been modified.

  1. Pages 3-4, sections 2.1, 2.2 and 2.3, discuss only the beneficial aspects of cotton rotations with other crops and lack some of the shortcomings of rotations.

Reply: They have been added.

  1. Page 5,lines 5-6“Therefore,crop rotation is a fundamental strategy to control residue-borne plant diseases.”should be changed to “Crop rotation can also be regarded as a strategy to control residual plant diseases.”

Reply: The sentence has been changed.

  1. Page 2-5. This section describes the advantages of crop rotation productivity, economic benefits, resource use, soil properties, root growth and biodiversity, but does not seem to be linked to root-shoot relationships.

Reply: Thank you for your comments. The root-shoot relationships under rotation has been

added.

  1. The title is mainly related to the root cap. It is suggested to adjust the order of the subtitles and contents (The same is true of the following two parts).

Reply: Agreed. The order of the subtitles and contents have been adjusted.

Cotton-based Intercropping

  1. Page 6, section 3.2, description of competitive relationships between cotton and other intercrops is lacking.

Reply: Since interaction includes competitive relationships, we do not specifically describe

the competitive relationship.

  1. Page 6, section 3.2, Intercropping improves the physicochemical properties of the soil,

which is mentioned in the section on crop rotation and is not mentioned here.

Reply: Good suggestion. Although both intercropping and rotation can affect the physical

and chemical properties of soil, we pay different attention to the two systems. In the intercropping, we paid more attention to rhizosphere microbial community and roots, because this is the current research hotspot and deserves attention.

  1. Page 6,lines 30-32. In this sentence “Compared with traditional cotton monocropping,

cotton–halophytes intercropping increases root mass and density at the 0−20cm soil depth”, “0−20-cm” Check whether it is correct.

Reply: After checking, we believe that 0−20cm was correct.

  1. Page 7, line 1, intercropping can alter not only the root secretions of the crop but also

the root structure and growth of the crop, which is not mentioned here.

Reply: Root structure and growth of the crop have been included.

  1. Page 7, line 24, intercropping can change the microclimate environment and will increase the number of beneficial insects and thus reduce the number of pests, not mentioned here.

Reply: These contents have been highlighted in red in the text.

  1. Page 7, line 24. Note the relationship between punctuation and spaces.

Reply: We apologize for this careless mistake. The sentence has been modified.

  1. Page 6-7, 3.2 has a lot of content, but 3.3 has little content. It is not written around the

main content of the article.

Reply: In this section, we mainly describe the impact of intercropping on the belowground

parameters including rhizosphere microorganisms, root exudates and root traits, which is consistent with the topic of this article. We also discussed the impact of intercropping on root traits to enrich the contents.

How Alternate Intercropping Improves Crop Productivity

  1. On page 7, "How Alternate Intercropping Improves Crop Productivity" talks about the

role of "Alternate Intercropping", which is inconsistent with the text, which mostly talks about the results associated with intercropping cotton with other crops and does not cover the "Alternate Intercropping" aspect.

Reply: We discussed how alternate intercropping improves crop productivity in the whole

paragraph on page 7 as shown below.

"The average land equivalent ratio can increase by 20% to 30% compared with that in

traditional intercropping. In a cotton–peanut intercropping system, traditional intercropping increases seed cotton yield by 16.9% and decreases peanut yield by 5.6%, whereas alternate intercropping increases cotton yield by 21% without sacrificing peanut yield. Therefore, the crop output value under alternate intercropping was 4.5% higher than that under traditional intercropping, and the net return exceeded that under traditional intercropping by 10% [2]. In maize and peanut alternate intercropping, annual yields of maize and peanuts increased by 19.68% and 17.29% and net revenues increased by 23.14% and 13.99%, respectively, compared with those in traditional intercropping [84]. Thus, compared with traditional intercropping, alternate intercropping increases crop productivity and economic return without additional inputs.

  1. Page 7,lines 29-31.“Alternate intercropping, or transposition intercropping, is a new intercropping mode in which two crops are intercropped in a wide strip with planting positions switched annually on the same land [2,84].” Check whether the references are correct.

Reply: The references are correct.

  1. Page 8, line 12, the text mentions cotton-peanut intercropping compared to traditional intercropping and does not raise which crop the traditional intercropping is.

Reply: We have rewritten the sentence.

  1. Page 8,lines 17-18.“In maize–peanut alternate intercropping.” The symbol in the sentence is suggested to be deleted.

Reply: Agreed. The symbol in the sentence has been deleted.

  1. Page 8, section 4.1, most of the text refers to the effect of cotton intercropping with a single crop on yield and productivity, with no mention of the rotation associated with cotton intercropping systems, which is inconsistent with the title.

Reply: The limitation you indicated does exist. However, there is very little literature on this aspect. Nevertheless, we will pay attention to the comparison between alternate cropping and rotation in the future.

  1. Page 9,line 6,“under maize–peanut alternative intercropping”This sentence is suggested to be changed to“under maize and peanut alternative intercropping”

Reply: Agreed. The sentence has been modified.

  1. Page 9,line 7,“SPAD”The first occurrence in the text should be the full name.

Reply: We apologize for this careless mistake. The full name has been added.

  1. Page 9,lines 11-12, it is suggested to add a title between the two paragraphs

Reply: Agreed. A title has been added.

  1. Page 9, line 33, format error.

Reply: We apologize for this careless mistake. The format has been modified.

  1. Page 9, line 33. Please note the formatting, the space before the paragraph.

Reply: The format has been modified.

  1. Page 10 line 16, it is suggested to put Figure 1 in the front to facilitate readers' understanding.

Reply: Agreed. The position of Figure 1 has been changed.

  1. Page10, line 18, format error.

Reply: The format has been corrected.

  1. Page 10, line 23, no mention of the advantages of "alternate intercropping" in relation to "intercropping".

Reply: The advantages of alternate intercropping over traditional intercropping have been clearly explained in parts 4.1, 4.2 and 4.3. Thus it is not necessary to discuss the advantages repeatedly. This part mainly discussed the aboveground and belowground interaction under alternate intercropping

  1. Page 10, line 18. Please note the formatting, the space before the paragraph.

Reply: The format has been modified.

  1. Page 11, line 18. Please note the formatting, the space before the paragraph.

Reply: The format has been modified.

  1. Page 12,line 4.“Arabidopsis ELONGATED HYPOCOTYL5 (HY5)”Check whether are

correct.

Reply: It is correct.

  1. Page 12 line 14, no punctuation at the end of the sentence.

Reply: We apologize for this careless mistake. The punctuation has been added.

 Discussion and Conclusions

  1. Page 12, line 20-25. Please note that the format of the references is the same as before.

Reply: The format of the references has been modified.

  1. Page 12, line 25, discussion of "alternate intercropping" is lacking.

Reply: The discussion of "alternate intercropping" was added.

  1. Page 12, line 32, missing punctuation.

Reply: The punctuation has been added.

  1. Page 13, line 7. Please note the correct use of punctuation at the end of sentences.

Reply: The sentence has been modified.

  1. It is suggested to add the limitations of discovery, which can be avoided in future exploration.

Reply: It has been added.

  1. A discussion of crop rotation patterns and years under intercropping systems in cotton is also needed and is not covered in the paper.

Reply: The pattern and years of rotation under alternate intercropping have been concisely discussed in the first paragraph of Section 4 ("intercropped and rotated with legumes like peanut, soybean and mung bean or grain crops like maize, sorghum and millet in a wide strip with planting positions switched annually on the same land"). We think the current description is enough, because these are not the main contents that we focus on.

 References

  1. DOI numbers are uniformly not added, or uniformly added, and the format should be uniform.

Reply: We apologize for this careless mistake. The format of references has been uniformed.

  1. Uniform abbreviations or uniform non-abbreviations for reference journals, please.

Reply: Agreed and changed.

  1. The format of all references cited is changed to corner mark.

Reply: Changes were made as the journal requested.

  1. According to the journal requirements, the journal in the reference needs to be abbreviated, so please revise the issue similar to the case of reference 2, 3, etc.

Reply: Agreed and revised.

Reviewer 2 Report

The paper by Lv et al is a straightforward and reasonably thorough review of research on cotton interacting with other crops through rotations, intercropping, and especially the combination of the two in “transposition” or “alternate” intercropping.  The paper is well-organized and generally well-written, though there is a need for revision of some grammatical and stylistic errors.  The wide range of literature cited attests to the advantages of these practices, and it is useful to have a compilation and synthesis of these studies.  The authors also provide a simple conceptual model with three types of relevant interactions, presented in a clear and attractive diagram.

However, I do feel serious revision is necessary regarding the use and development of the concept of “rebalancing” root and shoot relations, a term used repeatedly throughout the manuscript to describe a process that is said to occur when cotton is rotated, intercropped, or both.  Though mentioned extensively to the point of redundancy (see Section 4.3.2, top of p. 11), the term is never clearly defined, and the idea of what exactly occurs is vague and not supported by detailed explanation or data.  General comments are made on the canopy being modified which changes the rhizosphere, for example, or signaling occurring between root and shoot, and that alternate intercropping improves nutrient uptake or productivity because of this rebalancing.  But specific examples on, say, changes in nitrogen or phosphorus transport between root and shoot, or change in biomass or branching of roots under intercropping need to be described, preferably with tables or figures based on cited work.  As is, that there should be some reallocation of nutrients, photosynthate, or biomass with any number of environmental change is trivial without more information.  This is not to say that the data is not there, only that it needs to be presented explicitly.

In addition, I would mention the following points. (the manuscript I downloaded does not have line numbers):

Sec. 2.4.  Spiders are fewer in intercropped cotton, and it is implied that this is a positive thing.  Spiders are carnivores and generally act as beneficial predators of pests.  Could the authors mean spider mites?

Sec. 3. Intercropping is not done in only rows or bands.  For example, seed can be mixed and broadcast; in East Africa bean and maize seed are placed in the same planting hole.

Sec. 3.4.  Disease in cotton intercrops is not addressed.  Work has been done on this; see Boudreau et al (2016) Plant Pathology 65:601-611.

Sec. 4 Intro. Specify where transposition intercropping is “used widely.”  Intercropping, much less transposition intercropping is very rare in the U.S.

Sec. 4.1.  Care should be taken to generalize from few studies.  For example, on page 8, the authors write that “intercropping increases seed cotton yield by 16.9% …” and “Alternate intercropping provides distinct advantages underground.” These are categorical statements but only based on single studies.  Results should be qualified with “may” or “have been shown to” rather than blanket statements. 

Sec. 4.3.3. Most of this discussion does not describe results of intercropping or rotation, just general information on signaling.  It should be related more explicitly to these practices.

Sec. 5.  The Discussion and Conclusions section is not as well-written as the rest of the paper, and should be reviewed and edited.  For example, Sec. 5.2 ends with an incomplete sentence, and the first paragraph of Sec. 5.3 is very poorly written.

References.  DOIs are not consistently given.  Note error on Reference # 37.

Author Response

Reviewer #2

The paper by Lv et al is a straightforward and reasonably thorough review of research on cotton interacting with other crops through rotations, intercropping, and especially the combination of the two in “transposition” or “alternate” intercropping.  The paper is well-organized and generally well-written, though there is a need for revision of some grammatical and stylistic errors.  The wide range of literature cited attests to the advantages of these practices, and it is useful to have a compilation and synthesis of these studies.  The authors also provide a simple conceptual model with three types of relevant interactions, presented in a clear and attractive diagram.

However, I do feel serious revision is necessary regarding the use and development of the concept of “rebalancing”root and shoot relations, a term used repeatedly throughout the manuscript to describe a process that is said to occur when cotton is rotated, intercropped, or both.  Though mentioned extensively to the point of redundancy (see Section 4.3.2, top of p. 11), the term is never clearly defined, and the idea of what exactly occurs is vague and not supported by detailed explanation or data. General comments are made on the canopy being modified which changes the rhizosphere, for example, or #1 occurring between root and shoot, and that alternate intercropping improves nutrient uptake or productivity because of this rebalancing. But specific examples on, say, changes in nitrogen or phosphorus transport between root and shoot, or change in biomass or branching of roots under intercropping need to be described, preferably with tables or figures based on cited work. As is, that there should be some reallocation of nutrients, photosynthate, or biomass with any number of environmental change is trivial without more information. This is not to say that the data is not there, only that it needs to be presented explicitly.

Reply: Good comments. Thank you for your suggestion. We have included a definition of re-balancing in the text (4.4.2). However, due to the limited research reports available, no detailed information is provided in this paper. This will be the focus of our future attention and research.

In addition, I would mention the following points. (the manuscript I downloaded does not have line numbers):

Sec. 2.4.  Spiders are fewer in intercropped cotton, and it is implied that this is a positive thing.  Spiders are carnivores and generally act as beneficial predators of pests.  Could the authors mean spider mites?

Reply: In the literature, the spiders refer to wild spiders as described below:

 "However, primary and secondary (potential species) pest species for cotton and wheat in weeds growing around fields and along streams are particularly diverse entomophagous (parasitic, wild spiders, wild caterpillars, lizards, two-winged entomophagous, wild thrips) mainly in these areas". Therefore, wild spiders are pests rather than beneficial predators of pests.

Sec. 3. Intercropping is not done in only rows or bands.  For example, seed can be mixed and broadcast; in East Africa bean and maize seed are placed in the same planting hole.

Reply: Although intercropping can be conducted by mixed seeding and broadcast, it only refer to planting at least two crops in the same season in rows or bands in this paper.

Sec. 3.4.  Disease in cotton intercrops is not addressed.  Work has been done on this; see Boudreau et al (2016) Plant Pathology 65:601-611.

Reply: Agreed. We have referred to the suggested reference and added a discussion on control of disease with intercropping.

Sec. 4 Intro. Specify where transposition intercropping is “used widely.”  Intercropping, much less transposition intercropping is very rare in the U.S.

Reply: “widely” was deleted.

Sec. 4.1.  Care should be taken to generalize from few studies.  For example, on page 8, the authors write that “intercropping increases seed cotton yield by 16.9% …” and “Alternate intercropping provides distinct advantages underground.” These are categorical statements but only based on single studies.  Results should be qualified with “may” or “have been shown to” rather than blanket statements.

Reply: Good suggestion. We have modified the relevant statements.

Sec. 4.3.3. Most of this discussion does not describe results of intercropping or rotation, just general information on signaling. It should be related more explicitly to these practices.

Reply: We understand you concern. However, there are few studies on root-shoot signals under intercropping or rotation at present. So only general information on signaling was discussed in the text. We believe there must be signal transduction in intercropping and rotation systems, and these signals play a very important role, which is worth studying and discussing in the future. We will pay attention to these issues in our future research.

Sec. 5.  The Discussion and Conclusions section is not as well-written as the rest of the paper, and should be reviewed and edited.  For example, Sec. 5.2 ends with an incomplete sentence, and the first paragraph of Sec. 5.3 is very poorly written.

Reply: The Discussion and Conclusions section have been modified.

References.  DOIs are not consistently given. Note error on Reference # 37.

Reply: References have been modified.

Reviewer 3 Report

In this study, authors present “Cotton-based rotation, intercropping and alternate intercropping increase yields by improving root–shoot relations”. The study is valid to update the field & knowledge. However, this manuscript is Accepted with minor revisions for possible publication.

1.     Abstract is not comprehensive.

2.     There is a dire need to refine the hypothesis.

3.     In 2.1. Productivity and economic benefits: The author mentioned “compared with continuous cropping of cotton, rotation cropping increases cotton yields by 4.5% with rotary tillage and 2.6% with deep tillage”. What do you mean by deep tillage? Please define it.

4.     In section 2.4. Pest and disease control author suggested that crop rotation would be helpful in pest and disease control. Do you think it is? It may increase pest population if the pest can feed on both crops. some pests attack multiple crops so it would work or not?

5.     Author should recommend crop rotation pattern for top cotton growing countries.

6.     Author should elaborate on how the rotation will work under a climate change scenario. Please read and include (https://doi.org/10.3389/fpls.2021.727835; 10.3390/agronomy11091885; 10.3390/agronomy12061310)

7.     Does crop rotation will work under abiotic stress conditions like salt, heat and drought, etc?

8.     Language of the manuscript needs minor revisions.

Author Response

Reviewer #3

In this study, authors present “Cotton-based rotation, intercropping and alternate intercropping increase yields by improving root–shoot relations”. The study is valid to update the field & knowledge. However, this manuscript is Accepted with minor revisions for possible publication.

  1. Abstract is not comprehensive.

Reply: It has been revised to be more comprehensive.

  1. There is a dire need to refine the hypothesis.

Reply: This is a review article, and thus no hypothesis was included.

  1. In section 2.4. Pest and disease control author suggested that crop rotation would be helpful in pest and disease control. Do you think it is? It may increase pest population if the pest can feed on both crops. some pests attack multiple crops so it would work or not?

Reply: Good question. Intercropping may increase pest population as you indicated, but it can also increase the natural enemy, and thus the damage will not be more serious than monoculture in most cases. In some intercropping pattern, pest and disease are greatly reduced. Therefore, it is necessary to select the appropriate intercropping pattern that is helpful in pest and disease control.

  1. Author should recommend crop rotation pattern for top cotton growing countries.

Reply: Peanut and soybean are good crops to rotate with cotton.

  1. Author should elaborate on how the rotation will work under a climate change scenario. Please read and include (https://doi.org/10.3389/fpls.2021.727835; 10.3390/agronomy 11091885; 10.3390/agronomy12061310) 

Reply: The reference has been referred and cited. The following content has been included in the text.

Selecting crops with strong abotic resistance such as cotton and sorghum for rotation or intercropping can alleviate salinity damage. Developing shade-tolerant cotton varieties and adoption of cotton-fruit intercropping can alleviate the stress of high temperature on cotton to a certain extent. Under limited water or nitrogen fertilizer input, the temperature and precipitation in a year vary greatly. In this case, it is necessary to properly adjust the intercropping time, strip width and crop variety. Crop model can be adopted to manage and mitigate climate risk of under intercropping.

  1. Does crop rotation will work under abiotic stress conditions like salt, heat and drought, etc? 

Reply: Yes. Cotton-based rotation is helpful to reduce soil salinity.

  1. Language of the manuscript needs minor revisions.

Reply: We have made revisions to the text.

Reviewer 4 Report

Reviewer Comments:

 In this review titled “Cotton-based rotation, intercropping and alternate intercropping increase yields by improving root– shoot relations” different cotton-based planting patterns are analyzed to explain how the rebalancing of root–shoot relations ultimately affects yields. The review examines the effects of above- and belowground interactions and rebalancing of root–shoot relations on crop yields under intercropping, rotation, and particularly alternate intercropping with the practices combined. More related discussion should be included in the introduction section. The review is interesting and therefore, I would like to recommend the publication of this paper in the journal of Agronomy.

 Comments:

1.      Although Figure 1 is good and informative but authors must add more related figures to enhance the importance of this work?

2.      Authors should also add some comparison tables of different cotton-based planting patterns?

3.      As “5.1. Modeling on intercropping and rotation” is important topic I suggest to explore it more for enrichment of this draft as this manuscript lack figures and tables?

4.      Heading 5.1, 5.2 and 5.3 are without references.  Had nobody worked on it?

Author Response

Reviewer #4

 In this review titled “Cotton-based rotation, intercropping and alternate intercropping increase yields by improving root– shoot relations” different cotton-based planting patterns are analyzed to explain how the rebalancing of root–shoot relations ultimately affects yields. The review examines the effects of above- and belowground interactions and rebalancing of root–shoot relations on crop yields under intercropping, rotation, and particularly alternate intercropping with the practices combined. More related discussion should be included in the introduction section. The review is interesting and therefore, I would like to recommend the publication of this paper in the journal of Agronomy. 

 Comments:

  1. Although Figure 1 is good and informative but authors must add more related figures to enhance the importance of this work?

Reply: Thank you for your suggestion. We added Figure 1 to further show the patterns of cropping system.

  1. Authors should also add some comparison tables of different cotton-based planting patterns?

Reply: Good suggestion. However, we did not make accurate comparison among cotton-based cropping patterns due to difficulties in literature retrieval caused by the COVID-19 at present.

  1. As“5.1. Modeling on intercropping and rotation” is important topic I suggest to explore it more for enrichment of this draft as this manuscript lack figures and tables?

Reply: We added figures1 and enriched the part of 5.1. Modeling on intercropping and rotation.

  1. Heading 5.1, 5.2 and 5.3 are without references.  Had nobody worked on it?

Reply: We have added references. 

Round 2

Reviewer 1 Report

We have no more Suggestion